# DNA Binding and Anticancer Properties of New Pd(II)-Phosphorus Schiff Base Metal Complexes

**DOI:** 10.3390/pharmaceutics14112409

**Published:** 2022-11-08

**Authors:** Burcu Saygıdeğer Demir, Simay İnce, Mustafa Kemal Yilmaz, Aycan Sezan, Ezgi Derinöz, Tugba Taskin-Tok, Yasemin Saygideger

**Affiliations:** 1Department of Biotechnology, Institute of Natural and Applied Sciences, Çukurova University, Adana 01330, Turkey; 2Department of Nanotechnology and Advanced Materials, Institute of Science and Technology, Mersin University, Mersin 33343, Turkey; 3Department of Chemistry, Science Faculty, Mersin University, Mersin 33343, Turkey; 4Department of Chemistry, Faculty of Arts and Science, Gaziantep University, Gaziantep 27410, Turkey; 5Department of Bioinformatics and Computational Biology, Institute of Health Sciences, Gaziantep University, Gaziantep 27410, Turkey; 6Department of Pulmonary, School of Medicine, Cukurova University, Adana 01330, Turkey; 7Department of Translational Medicine, Institute of Health Sciences, Çukurova University, Adana 01330, Turkey

**Keywords:** molecular docking, phosphorus Schiff base, metal complexes, cancer cells, antitumor, DNA binding, cytotoxicity, apoptosis

## Abstract

DNA has become the target of metal complexes in cancer drug discovery. Due to the side effects of widely known cisplatin and its derivative compounds, alternative metal-based drug discovery studies are still ongoing. In this study, the DNA-binding ability of Pd(II) and Pt(II) complexes of four phosphorus Schiff base ligands and four hydrazonoic-phosphines are investigated by using in silico analyses. Phosphorus Schiff base-Pd(II) complexes encoded as **B1** and **B2** with the best DNA-binding potential are synthesized and characterized. The DNA-binding potentials of these two new Pd(II) complexes are also investigated experimentally, and their antitumor properties are demonstrated in vitro in A549, MCF7, HuH7, and HCT116 cancer cells. The mechanisms of these metal complexes that kill the cells mentioned above in different activities are elucidated by flow cytometry apoptosis analysis and colony formation analysis The in silico binding energies of these two new palladium complexes ΔG (**B1**): −4.51 and ΔG (**B2**): −6.04 kcal/mol, and their experimental DNA-binding constants were found as Kb (**B1**): 4.24 × 10^5^, Kb (**B2**): 4.98 × 10^5^). The new complexes, which show different antitumor effects in different cells, are the least effective in HuH7 liver cells, while they showed the best antitumor properties in HCT116 colon cancer cells.

## 1. Introduction

A wide variety of cytotoxic drugs with different mechanisms of action have been discovered to treat cancer, which is among the deadliest diseases of the 21st century, and some of them are used alone or in combination to fight cancer [1]. The biggest disadvantage of these drugs is the side effects they cause due to the activities they show without distinguishing between cancer cells and healthy cells. To fight this disease, targeted drug discovery with fewer side effects is of great importance. After cisplatin’s discovery, various recent discovery studies have focused on organometallic compounds combining transition metals and their chelates [2,3].

There are important similarities between the coordination chemistry of palladium(II) and platinum(II) compounds. One of the most important factors in the preference of platinum compounds as metal-based antitumor agents is ligand exchange kinetics. Whereas the rate of hydrolysis in palladium complexes is 10^5^ times higher than the corresponding platinum analogues. Pd(II) makes the ligand to which it binds highly reactive in terms of reaching pharmacological targets [4]. Compared with cisplatin, the corresponding cispalladium, *cis-*[Pd(NH_3_)_2_Cl_2_] and *cis*-[Pd(DACH)Cl_2_] (DACH: (1R,2R)-(−)-1,2-diaminecyclohexane) show no antitumoral activity. It is well known that the former undergoes an inactive trans-conformation and both compounds interact in vivo with many molecules, particularly proteins, preventing them from reaching the pharmacological target DNA. This is evidence that palladium compounds hydrolyze very well [5,6,7]. Because of the extremely high activity of palladium complexes, if an antitumor palladium drug is to be developed, it must be somewhat stabilized by a strongly coordinated nitrogen ligand and an appropriate leaving group. One of the ways to stabilize palladium compounds and prevent any possible cis-trans isomerism is to synthesize Pd(II) complexes with bidentate ligands [8]. Pd(II) complexes with phosphine ligands [9,10] and Palladium(II) complexes with mixed donor atom ligands were preferred because Pd(II) compounds can target DNA as a stabilized structure, some of them have been shown to have better antitumor properties and are less toxic than cisplatin [11,12,13].

In this study, Pd(II) and Pt(II) complexes of four different phosphorus Schiff base ligands and four different hydrazonoic-phosphines were designed and their binding energies to DNA were evaluated by the molecular docking method. The Pd(II) complexes of the two different phosphorus Schiff base ligands with the highest DNA-binding potential were synthesized and characterized. The antitumor activities and experimental DNA-binding capacities and anticancer activities of these two new palladium compounds were evaluated.

## 2. Materials and Methods

### 2.1. Materials

Ligand and complex synthesis were carried out under argon atmosphere using standard Schlenk techniques. Dichloromethane (CaH_2_, 5% *w*/*v*) and toluene (CaH_2_, 5% *w*/*v*) were dried and distilled prior to use. The palladium precursor dichloro(1,5-cyclooctadiene)palladium(II) (Pd(cod)Cl_2_) and 3′-aminoacetophenone were purchased from Sigma-Aldrich and used as received. The ^1^H, ^31^P, and ^13^C NMR analyses were recorded on Bruker Avance 400 spectrometer and the chemical shifts (δ) were expressed in ppm relative to Me_4_Si as internal standard at 400.2, 162.0, and 100.6 MHz, respectively. Spin multiplicities are designated by the following symbols: s, singlet; d, doublet; t, triplet; dd, doublet of doublets; ddd, doublet of doublet of doublets with coupling constants (J, Hz) or m, multiplet.

Four different cancer cell lines including lung non-small cell lung carcinoma A549 (ATTC, CCL-185, Washington, DC, USA), breast cancer MCF7 (ATCC, HTB-22), hepatocellular carcinoma cell line HuH7 (NIBIOHN, JCRB0404) (Cell Bank, Tokyo, Japan, JCRB0403, CVCL_0336) and colorectal cancer cell line HCT116 (ATCC, CCL-247). and one transformed normal lung epithelial cell line BEAS2B (RRID:CVCL_0168) (ATTC, Washington, DC, USA, CRL-9609) used to evaluate anticancer activities of novel complexes. A549 and BEAS2B cell lines were provided from Prof. H. Bayram, Gaziantep University, HuH7 from Prof. M. Ozturk, IBG Izmir, MCF7 and HCT116 were by Prof. M.B. Yilmaz Cukurova University, Adana. All cells were maintained in RPMI 1640 (HyClone Cat No:SH30027.01) supplemented with 10% fetal bovine serum (HyClone Cat No: SV30160.03) and 1% antibiotics (HyClone SV30079.01), in CO_2_ incubator (Thermo Sci.) at 5% C = 2 and 21% oxygen. Cells were harvested using Tyripsin (HyClone Cat. No: SH30042.01) and PBS 1X (HyClone Cat. No: SH30256.01) was used to wash the plated cells or cell pellets when necessary. Complexes were dissolved in DMSO (Sigma Aldrich, St. Louis, MO, USA) at 10 mM stock concentration and further diluted in cell culture media during the experiments.

### 2.2. Molecular Docking Analysis

The chemical structures of the designed Pd(II) and Pt(II) complexes (Figure 1) as ligands were drawn and then geometry optimization was made at MO-G/PM6 level by using SCIGRESS [14], as shown in Appendix A. The crystal structure of DNA (PDB ID: 1BNA), was obtained from the protein data bank [15]. The target was also minimized CHARMm force field and the adopted-basis Newton-Raphson (ABNR) method [16] using DS 3.5 software [17] to prepare the target for docking process. After preparation of target and ligand(s), Autodock 4.2 [18] was performed to conduct the docking analysis on the basis of *cisplatin* and *triphenylphosphine* for DNA as control compounds.

### 2.3. Synthesis of Phosphorus Schiff Base Ligands (**L1** and **L2**)

**L1:** To a solution of 2-diphenylphosphinobenzaldehyde (1.000 g, 3.4 mmol) in dry toluene (10 mL) was added 3′-aminoacetophenone (0.470 g, 3.5 mmol) under argon and the solution was stirred at reflux temperature overnight. After this time the solvent was evaporated to dryness and then the residue was dissolved in absolute ethanol and crystalized at −20 °C., to give pure **L1.** Yield: 1.34 g, %97. White solid. **^1^H NMR (400.2 MHz, CDCl_3_):** δ (ppm) 8.98 (d, J_PH_ = 5.1 Hz, 1H, CH=N), 8.10 (ddd, J = 7.7, 3.8, 1.3 Hz, 1H), 7.71–7.67 (m, 1H), 7.42–7.37 (m, 1H), 7.36 (t, J = 1.8 Hz, 1H), 7.31–7.22 (m, 12H), 7.00 (ddd, J = 7.9, 2.1, 1.1 Hz, 1H), 6.87 (ddd, J = 7.7, 4.6, 1.0 Hz, 1H), 2.47 (s, 3H, CH_3_) (Appendix A). **^31^P NMR (162.0 MHz, CDCl_3_):** δ (ppm) −12.61 (s) (Appendix A).

**L2:** To a solution of 2-(di-p-tolylphosphino)benzaldehyde (1.080 g, 3.4 mmol) in dry toluene (10 mL) was added 3′-aminoacetophenone (0.470 g, 3.5 mmol) under argon and the solution was stirred at reflux temperature overnight. After this time the solvent was evaporated to dryness and then the residue was dissolved in absolute ethanol and crystalized at −20 °C, to give pure **L2.** Yield: 1.45 g, %98. White solid. **^1^H NMR (400.2 MHz, CDCl_3_):** δ (ppm) 9.07 (d, J_PH_ = 5.2 Hz, 1H, CH=N), 8.19 (ddd, J = 7.7, 3.8, 1.2 Hz, 1H), 7.78–7.73 (m, 1H), 7.45–7.42 (m, 2H), 7.40–7.34 (m, 2H), 7.23–7.14 (m, 8H), 7.09 (ddd, J = 7.9, 2.1, 1.0 Hz, 1H), 6.95 (ddd, J = 7.6, 4.7, 1.0 Hz, 1H), 2.53 (s, 3H, C(O)CH_3_), 2.35 (s, 6H, CH_3_) (Appendix A). **^31^P NMR (162.0 MHz, CDCl_3_):** δ (ppm) −14.73 (s) (Appendix A).

### 2.4. Synthesis of Pd(II) Complexes of Phosphorus Schiff Base Ligands (**B1** and **B2**)

**B1:** Pd(cod)Cl_2_ (0.142 g, 0.50 mmol) was dissolved in dry dichloromethane (5.0 mL). To this solution was added **L1** (0.290 g, 0.50 mmol) solution in dry dichloromethane (5.0 mL), and the reaction mixture was stirred for 10 min at room temperature, and the resulting precipitate was filtered off and dried under high vacuum, to obtain complex **B1**. Yield: 0.40 g, 93%. Pale yellow solid. **^1^H NMR (400.2 MHz, DMSO-d6):** δ (ppm) 8.76 (s, 1H, CH=N), 8.24 (dd, J = 6.9, 4.1 Hz, 1H), 8.00 (t, J = 7.6 Hz, 1H), 7.92 (d, J = 7.8 Hz, 1H), 7.88–7.83 (m, 2H), 7.75–7.70 (m, 2H), 7.65 (td, J = 7.5, 2.7 Hz, 5H), 7.55 (ddd, J = 8.5, 5.4, 1.4 Hz, 5H), 7.09 (dd, J = 10.4, 7.8 Hz, 1H), 2.61 (s, 3H, CH_3_) (Appendix A). **^31^P NMR (162.0 MHz, CDCl_3_):** δ (ppm) 31.05 (s) (Appendix A). **^13^C NMR (101.6 MHz, DMSO-d6)** *δ* (ppm) 197.0 (s, C=O), 168.6 (d, *J* = 8.3 Hz, C=N), 151.8 (s, ArC-N), 137.9 (d, *J* = 8.5 Hz), 136.7 (s), 136.4 (d, *J* = 15.0 Hz), 135.0 (d, *J* = 7.9 Hz), 133.8 (d, *J* = 11.2 Hz), 133.3 (s), 132.5 (d, *J* = 2.7 Hz), 129.2 (d, *J* = 11.8 Hz), 128.7 (s), 128.2 (s), 127.4 (s), 125.3 (s), 124.7 (s), 122.3 (s), 120.7 (s), 120.2 (s), 26.8 (s, C(O)CH_3_) (Appendix A).

**B2:** Complex **B2** was similarly prepared using **L2** (0.152 g, 0.50 mmol) and Pd(cod)Cl_2_ (0.29 g, 0.50 mmol). Yield: 0.42 g, %95. Yellow solid. **^1^H NMR (400.2 MHz, DMSO-d6):** δ (ppm) 8.71 (s, 1H, CH=N), 8.20 (dd, J = 7.5, 4.0 Hz, 1H), 7.95 (t, J = 7.6 Hz, 1H), 7.89 (d, J = 7.8 Hz, 1H), 7.80 (s, J = 16.3 Hz, 2H), 7.68 (d, J = 7.9 Hz, 1H), 7.54 (t, J = 7.8 Hz, 1H), 7.47–7.38 (m, 7H), 7.07 (dd, J = 10.4, 7.8 Hz, 1H), 2.59 (s, 3H, C(O)CH_3_), 2.39 (s, 6H, CH_3_) (Appendix A). **^31^P NMR (162.0 MHz, CDCl_3_):** δ (ppm) 30.18 (s) (Appendix A). **^13^C NMR (101.6 MHz, DMSO-d6)** *δ* (ppm) 197.0 (s, C=O), 168.4 (d, *J_PC_* = 8.3 Hz, C=N), 151.7 (s, ArC-N), 142.8 (d, *J_PC_* = 2.9 Hz, ArC-CH_3_), 137.8 (d, *J_PC_* = 8.4 Hz), 136.7 (s), 136.3 (d, *J_PC_* = 14.9 Hz), 134.9 (d, *J_PC_* = 8.0 Hz), 133.7 (d, *J_PC_* = 11.6 Hz), 133.1 (s), 129.8 (d, *J_PC_* = 12.2 Hz), 128.7 (s), 128.3 (s), 127.4 (s), 26.8 (s, C(O)CH_3_), 21.1 (s, CH_3_) (Appendix A).

### 2.5. Cytotoxicity Assay

The novel metal complexes were tested for their cytotoxicity against A549 (lung), MCF7 (breast), HUH7 (hepatocellular), and HCT116 (colon) cancer cell lines and BEAS2B normal lung cells as mentioned above, using MTT (Thiazolyl Blue Tetrazolium Bromide) method according to the literature [19]. Briefly, the cells (1.0 × 10^4^/200 µL/well) were cultured in a 96-well plate overnight at 37 °C, 5% CO_2_ in their respective medium containing 10% FBS and 1% antibiotics. After 24 h, old medium was removed and the cells were incubated with 0–300 µM concentrations of the compounds for 24 h at 37 °C, 5% CO_2_. Cells with 0.1% DMSO (vehicle control) were also incubated at the same conditions. Cisplatin was also tested at the same condition as a positive control. After incubations, 10 μL of MTT solution (5 mg/mL in PBS buffer) was added and the cells were further incubated at 37 °C, 5% CO_2_ for extra 4 h to metabolize MTT by viable cells. After MTT treatment, the supernatants were carefully removed, 50 μL DMSO was added to each well and then absorbance was measured at 570 nm in a microplate reader (Biochrom EZ Read 400).

### 2.6. Apoptosis Assay

Annexin-V staining was performed according to the protocol of Biolegend apoptosis detection kit [20]. For quantitative analysis, 100 µM concentrations of metal complexes were tested on A549, MCF7, HUH7, and HCT116 cells. Cancer cells (1.0 × 10^5^ cells/mL) suspension in serum-free medium was incubated with the respective compound in 6-well plates in a CO_2_ incubator. After treatment with compounds for 48hrs, the cancer cells were harvested and incubated with APC-Annexin V and PI. The fluorescence emission of APC- Annexin-V-stained cells was measured at 633 nm (Red laser) in a flow cytometer (Beckman Coulter/CytoFLEX, Indianapolis, IN, USA). Dots represent cells as follows: lower left quadrant, living cells (APC−/PI−); lower right quadrant, early apoptotic cells (APC+/PI−); upper left quadrant, necrotic cells (APC−/PI+); upper right quadrant, late apoptotic cells (APC+/PI+).

### 2.7. Colony Forming Efficiency Assay

Colony forming efficiency assay was performed in 6-well plates. The cells were seeded at a density of 200 cells/well (for MCF7 and HUH 7) and of 300 cells/well (for A549 and HCT116) in 3 mL complete culture medium. Two different concentrations of the metal complexes were applied to the cells for 24 h. Then the old medium was removed, and new fresh complete medium (not containing metal complexes) was added to the cells. The medium was changed with the new one at twice in a week. Totally after two weeks, the colonies formed were dyed with methylene blue solution (50% methanol, 50% distilled water, and 0.4 g methylene blue) and were counted using ImageJ 1.53a software (NIH, USA). The differences between the groups were analyzed with Prism v8 (GraphPad, Inc., San Diego, CA, USA).

### 2.8. BSA Binding Assay

To examine the protein binding kinetics of compounds, 1 µM BSA was dissolved in PBS (PH: 7.0) and the change in absorbance was monitored in UV-visible spectrophotometer at 200–400 nm wavelength by increasing the concentration of the compounds (0–10 µM) (UV titration) against the constant amount of BSA.

### 2.9. DNA Binding Assay

All experiments involving the binding of **B1** and **B2** complexes with CT-DNA, 1mM Tris-HCl with 1 mM EDTA, pH 7.5 was carried out within. The 1 mM CT-DNA solution in this buffer gave a UV absorption peak (single) of 2.55 at 260 and 280 nm, indicating that the DNA was pure [21]. DNA concentration per nucleotide was determined by absorption spectroscopy using a molar extinction coefficient value of 6600 dm^3^ mol^−1^ cm^−1^ at 260 nm [22]. Compounds were dissolved in a solvent consisting of 5% DMSO and 95% pH 7.5 Tris buffer. With constant concentrations of compounds (25 µM), the DNA concentration was gradually increased (2.5–25 µM) and absorption titration was performed. An equal amount of DNA was added to both the test solution and the reference solution to eliminate the DNA self-absorption at the wavelength of study. Absorption titrations were calculated by employing the Wolfe–Shimmer equation [23]: [DNA]/(|εA − εF|) = [DNA]/(|εB − εF|) + 1/{Kb (|εB − εF|)}, where [DNA] is the concentration of DNA in base pairs; εA, εF, and εB correspond to Aobsd/[compound], the extinction coefficient of the free complex and the extinction coefficient of the compound in the fully bound form, respectively. In the plot of [DNA]/(|εA − εF|) versus [DNA], the intrinsic binding constant Kb is then given by the ratio of the slope to the intercept.

### 2.10. Statistical Analysis

Statistical analyzes were performed with Prism V.8 Software. (Graph-Pad, USA). Nonlinear regression analysis was performed for IC_50_ calculations in the cytotoxicity test, and Pearson chi-square tests were used to compare two different conditions. *p* < 0.05 was considered significant.

## 3. Results and Discussion

### 3.1. Palladium Complexes **B1** and **B2** Had the Most Potent Interaction with DNA According to In-Silico Analysis

Molecular docking was used to evaluate the structure-activity relationship between DNA and eight newly developed Pd and Pt complexes. Based on molecular docking results, the binding sites, and interactions of two compounds (**B1** and **B2**) with enhanced DNA activity were examined. B5 to B8 had poor DNA interaction. Table 1 shows all complexes’ binding energies under these conditions, while Figure 1 shows solely **B1** and **B2′**s three-dimensional interactions with DNA. The Appendix A includes the other structure interactions.

The binding energy values of **B1**–**B4** show a very moderate binding tendency. Accordingly, the inhibition constants also indirectly show less inhibitory properties. In particular, the closeness between **B2** and **B1** binding values and Ki values is closer and more related than the others.

On the other hand, the interaction of **B5** to **B8** complexes with DNA exhibits positive binding in Table 1. Because the related structures create unfavorable bumps with the target, that is, steric interactions, they realize an unstable state with DNA that is very weak and not supported by positive thermodynamics. These interactions are described in the Appendix A and also in the 2D and 3D interaction images in Appendix A.

Based on the binding affinities and inhibition constants of eight compounds and two control compounds with the target structure DNA, the atomic level situations of **B1** and **B2** had the highest binding affinity (Table 1). Therefore, B1 and B2 were selected for further evaluation in in vitro studies. Cisplatin creates hydrogen bonds with the A: DA5 and A: DA6 nucleotides of the target model, as shown in Appendix A. As shown in Appendix A, the triphenylphosphine molecule forms hydrogen bonds with the A: DA6 nucleotide and hydrophobic interactions with the A: DA5, A: DA6, and B: DA17 nucleotides of the target. In addition, due to its higher surface area, the triphenylphosphine structure, which was one of the control compounds, interacted with DNA more than cisplatin. As a result of these observations, the second control drug, triphenylphosphine (BE: −4.02 kcal/mol; Ki: 1.14 × 10^6^ nM), displayed a higher affinity for target binding than cisplatin (BE: 2.73 kcal/mol; Ki: 9.98 × 10^6^ nM).

Among the Pd complexes in the study, **B1** and **B2** exhibit dominant biological activity against DNA. They even outperformed control compounds in terms of DNA-binding affinity. Because of the methyl placement of the para positions in two phenyl rings of the phosphine group, **B2** performed the best docking and biological activity toward DNA, with a binding energy value of −6.04 kcal/mol and an inhibition constant of 3.73 × 10^6^ nM. Furthermore, as shown in Figure 1, **B2** formed one hydrogen bond with B: DA17 nucleotide, a pi–pi stacked interaction with A: DA5, three pi–pi-T-shaped interactions with A: DA5, A: DT7, and B: DA17 nucleotides, and two pi-alkyl interactions with A: DG4 and A: DA5 nucleotides of DNA. Detailed information between **B2** and DNA binding is given in Appendix A.

**B1**, on the other hand, had a more hydrophilic structure than **B2**. As a result, B1, binding energy, and inhibition constant data of −4.51 kcal/mol and 4.91 × 10^6^ nM were found to be lower when compared to the target model. The **B1** has two H-bonds with A: DA6 and B: DA18, and one hydrophobic interaction with the B: DA17 nucleotide of DNA. Figure 1 also summarizes this situation in three dimensions. Despite the active compounds, B3 and B4 differ structurally from **B1** and **B2** in that their chemical structure contains Pt metal rather than Pd metal. The presence of this metal reduces the expressed compounds’ DNA-binding affinity (Table 1). The difference in docking orientation of B3 and B4 structures in active regions of DNA and the low reactivity of the Pt metal in their chemical structures compared to the Pd metal are the reasons for this. Appendix A show the three-dimensional interactions of B3 and B4 with DNA. Appendix A shows the different types of interactions.

### 3.2. Phosphorus Schiff Base Ligands and Palladium(II) Complexes Were Successfully Synthesized and Characterized

The **L1** and **L2** ligands were prepared from the classical condensation reaction of 2-diphenylphosphinobenzaldehyde and 2-(di-p-tolylphosphino) benzaldehyde with 3′-aminoacetophenone**,** respectively. The reaction of corresponding phosphorus Schiff base ligands with Pd(cod)Cl_2_ in dichloromethane gave Pd(II) complexes (**B1** and **B2**) in high yields, as shown in Figure 2. Both complexes have been found to be stable for several weeks, and they are poorly soluble in dichloromethane, methanol, and chloroform, but readily soluble in dimethyl sulfoxide.

The structural characterization of compounds was evaluated with ^1^H, ^31^P NMR, and ^13^C analyses. The ^1^H NMR spectra of ligands displayed characteristic doublet signals associated with the imine hydrogen coupled with the phosphorus atom at *δ* 8.98 (d, *J_PH_* = 5.1 Hz, 1H, CH=N) and 9.07 ppm (d, *J_PH_* = 5.2 Hz, 1H, CH=N), respectively. The appearance of the singlet signals of the same protons in Pd(II) complexes at *δ* 8.76 (**B1**) and 8.71 ppm (**B2**) confirmed the coordination of phosphorus Schiff bases to the palladium center. The singlet signal for the methyl protons of the aceto group appeared at *δ* 2.61 (**B1**) and 2.59 ppm (**B2**), whereas the methyl groups attached to position 4 of the phenyl rings in **B2** appeared at *δ* 2.39 ppm (s, 6H, CH_3_). The formation of phosphorus Schiff base-palladium(II) complexes was also confirmed by ^31^P NMR analysis. The signals observed at *δ* −12.61 for ligand **L1** and *δ* −14.73 ppm for ligand **L2** disappeared and low-field singlet signals have newly formed at *δ* 31.05 for complex **B1** and *δ* 30.18 ppm for complex **B2,** which further supported the coordination of phosphorus donor to the palladium center. Thus, as expected, ^1^H and ^31^P NMR studies and ^31^C NMR spectroscopic data together support and are in agreement with the proposed structures (see Appendix A) [24,25,26].

### 3.3. **B1** and **B2** Showed Cytotoxic Activity on Cancer Cells

Pd(II)-phosphorus Schiff base complexes encoded as **B1** and **B2**, showed cytotoxic activity in all cancer cell lines (A549, MCF7, HUH7, and HCT116). A549 and HUH7 cells showed a not very good cytotoxicity, with IC_50_ values above 100 µM. Both **B1** (IC_50_ = 39.54 µM) and **B2** (IC_50_ = 42.02 µM) showed a good cytotoxic effect in MCF7 cells. Both were more effective than cisplatin (IC_50_ = 74.60 µM). In HCT 116 cells, **B2** showed better cytotoxic activity (IC_50_ = 24.82 µM) than **B1** (IC_50_ = 35.77 µM), and both showed better cytotoxic activity compared to cisplatin (IC_50_ = 37.70 µM) (Figure 2). The fact that the cytotoxic effect of **B2** is quite low in BEAS 2B normal lung cells, especially when compared to cancer cell lines, and **B1’s** better cytotoxic effect in BEAS2B normal lung cells compared to MCF7 and HCT116 cell lines, and that it may cause fewer side effects are promising. At the same time, while cisplatin showed an IC_50_ value of 21.40 µM in BEAS2B cells, much higher IC_50_ values of **B1** (92.86 µM) and **B2** (157.9 µM) were also recorded (Figure 2). In a previous in vitro anticancer study using Schiff base-metal complexes, IC_50_ values were found to be above 100 M and still considered cytotoxic [27]. In a study comparing the antitumor effects of Schiff base Pd(II) and Pt(II) complexes in MCF7 and HCT116 cells, it was discovered that the Pd(II) complex outperformed the free ligand and the Pt(II) complex containing the same ligand in the MCF7 cell line [28]. MCF7 and HCT116 cells produced the best results in our study. It is possible that Schiff base Pd(II) complexes will have a specific effect on MCF7 cells. While the cytotoxic effect in MCF7 was similar to ours but less effective (IC_50_ = 47.5 M), **B1** and **B2** in HCT116 cells demonstrated significantly higher efficacy (IC_50_ = 92.97 M) compared to the aforementioned study [28]. This and similar studies show how different results can be caused by the nature of the ligand as well as the type of metal in terms of cytotoxic activity.

### 3.4. **B1** and **B2** Induced Apoptosis in Cancer Cells

When the apoptotic cell death rates of cells treated with 100 µM **B1** and **B2** were evaluated separately, apoptotic cell death was found in the range of approximately 50–60% in A549 and HCT116 cells, with no significant difference between **B1** and **B2**. While the apoptotic activity of **B1** (63.73%) was higher than that of **B2** (39.09%) in MCF 7 cells, it was observed that the apoptotic activity of **B2** (83.51%) was higher than that of **B1** (47.14%) in HUH7 cells (The percentages given were early (lower right) and late (upper right) apoptotic cells) (Figure 3).

The increased apoptotic effect of **B1** in MCF7 cells and **B2** in HUH7 cells may be attributed to the differences in the types of these cancer cells and their biological behavior and needs further investigations. The only structural difference between **B1** and **B2** is a methyl functional group in **B2** attached to two phenyl rings in the ligand’s structure. This situation once again indicates how important the functional groups of the ligand are as well as the effectiveness of the metal in organometallic cancer drug research. In many metal-based drug studies, it is seen that the anticancer activity of different ligands belonging to the same metal is different from each other [29,30]. The difference in the intramolecular electronic nature also affects the interactions of the ligand with the metal and, therefore, the interactions with the biomolecule to which the metal complex is attached [31].

### 3.5. **B1** and **B2** Suppressed Colony Formation of Cancer Cells

In the colony formation assay, the effects of **B1** and **B2** complexes on the colonization of A549, MCF7, HUH7, and HCT116 cells were investigated. After two weeks, a significantly reduced colonization was observed in **B2**-treated A549 cells compared to an equal amount of DMSO-treated control (Figure 4). In MCF7 cells**, B1** suppressed colony formation at 100 µM and **B2** at 50 µM. Both **B1** and **B2** inhibited colony formation to almost the same degree at doses (120 µM) applied to HUH7 cells (Figure 4). The strong inhibitory effect of these two new palladium compounds on colony formation in the HCT116 cell is shown in Figure 5. Therefore, it is clear that these new Pd(II) complexes inhibit the colony-forming ability of cells, and these results are paralleled by cytotoxicity.

### 3.6. **B1** and **B2** Had Strong DNA Intercalating Activity

Titration followed by absorption spectroscopy is a powerful experimental approach to investigate the way metal complexes bind to DNA [32,33]. Any interaction between the complex and DNA is expected to change the spectral transitions in the organic part of the complex. Intercalation of small molecules with DNA results in more redshift and hypochromism [34]. In the case of groove bonding or electrostatic attraction between small molecules and DNA, there may be little or no redshift and hyperchromism reflecting some conformational changes. The **B1** and **B2** complexes exhibited an absorption band at a maximum of 233 and 236 nm, respectively. Upon gradual addition of CT DNA to **B1** and **B2**, hypochromism was observed and only a slight blueshift (3 nm) in **B2**, suggesting that both complexes are intercalated rather than groove-bonded and electrostatic interactions (Figure 6. Kb values were 4.24 × 10^5^ M^−1^ and 4.98 × 10^5^ M^−1^ for **B1** and **B2** complexes, respectively according to the [DNA]/(|εA − εF|) vs. [DNA] intersection ratio of the slope (Figure 6). When these values are compared with those of the classical EB-DNA intercalator (Kb = 1.4 × 10^5^ M^−1^), it can be said that these two new palladium complexes are strongly intercalating [35]. In a study with Zn(II), Pd(II), Cr(III), and VO(II)-Schiff base complexes containing the same ligand, it was determined that the most strongly binding compound to DNA was found both in in vitro ct-DNA-binding studies and in molecular docking analysis. It has been shown that it contains Pd(II), and its cytotoxic activity is better than the others [36]. This and many similar findings in the literature, as in our study, are evidence showing the ability of Pd(II) complexes to bind to DNA.

### 3.7. **B1** and **B2** Showed Static BSA Binding

It is very important to examine interactions with serum albumin, the most abundant protein in plasma, in drug research. As a result of binding to albumin protein, drug molecules may lose or improve their biological properties while being transported to desired targets in the body. Therefore, it is very important to reveal the mechanism of interaction of potential drugs with serum albumin. Since bovine serum albumin (BSA) is structurally the most similar to human serum (HSA), it is widely used in this type of research. BSA binding studies were performed using UV-vis spectroscopy. Changes in the UV spectrum of BSA caused by the aromatic amino acids tryptophan (Trp), tyrosine (Tyr), and phenylalanine (Phe) were determined by monitoring the absorption graphs at 280 nm. By comparing the characteristic absorption spectra of solutions of pure BSA and BSA-Pd(II) Schiff base complexes, it was determined whether the complexes bind to BSA statically or dynamically. In dynamic coupling, it only affects the excited state of the fluorophore group, so no change in the absorption spectra is observed. In static binding, the formation of a new BSA complex causes a change in the absorption spectrum of the fluorophore group. The gradual addition of **B1** and **B2** complexes to BSA leads to an increase in the intensity of the absorption band at 280 nm, supported by a blueshift of about 9 nm for **B1** and 8 nm for **B2** (Figure 6). According to this result, it can be deduced that the metal complexes change the polarity of the microenvironment around the Trp and Tyr amino acids of BSA. The change in the absorption band of BSA seen in the presence of **B1** and **B2** is evidence that the complex-BSA interaction is in the form of static binding [37,38,39].

## 4. Conclusions

It has been determined that the newly synthesized Pd(II)-phosphorus Schiff base complexes **B1** and **B2** have a strong binding ability to DNA both in silico and experimentally. These new compounds showed cytotoxic, apoptotic, and colonization inhibitory effects in different cancer cell lines and the results of these in vitro assays show that **B1** and **B2** have the potential to be identical or even better compared to cisplatin in in vitro trials.

## Data Availability

All the data will be provided by the authors of the manuscript when needed.

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
