# Peer review of "DNA Binding and Anticancer Properties of New Pd(II)-Phosphorus Schiff Base Metal Complexes"

_pharmaceutics, 2022, doi:10.3390/pharmaceutics14112409_

Round 1

Reviewer 1 Report

This manuscript is a very useful because it examines the effects of a new metal complex on DNA. However, this manuscript should be revised once before being accepted because of some problems.

---

Most of the problems are about the format. This manuscript is very tough to read.

Moreover, there are many variations in notation used in this draft. (e.g., compounds such as "B1" are written sometimes in Bold, sometimes in Normal, K of “Ki” should be written in italic).

I found the following typos,

l.19 unnatural spaces.

l.49 "in vivo" is in italic

l. 165 Section 1.2., 1.3.??

1. 181 106 nM => 106 nM

The link in l. 237 [28] is broken.

Also, I don't know what kind of calculation the authors used for the interaction energy with the metal complex.

If they used Autodock for the binding energy, the interaction energies are calculated for many conformations. I think the authors employed "the best score" among the conformations. If so, the authors should add such explanations for revised version.

The results of Table 1 are not clear. The correlation between the binding energy and Ki seems to be poor. The authors simply refer to the “positive binding energy” from B5 to B8 as "poor interaction", but I want to know any cause for this poor interaction. The authors should explain the numbers they obtained.

Section 3.6. suddenly mentions binding with BSA. Is this really relevant to the discussion?

Reviewer 2 Report

The paper entitled ‘DNA binding and anticancer properties of new Pd(II)-phosphorus Schiff base metal complexes’ provided an evidence that the new complexes showed antitumor effects in different cells. Overall, the paper is well prepared. However, some questions are presented below:

Major comments

1.     Authors presented that the complexes showed different antitumor effects in different cells. Thus, why the authors examined only one type of BEAS2B normal lung cells?

2.     The X-bar of Figure 2 should represent in term of concentration without ‘log’.

3.     Line 259-260, the references should be addressed and discussed for ‘the specific pathways in these cancer types is affected.

4.     The author should mention the % of DMSO-treated control as shown in line 272.

Minor comments

1.     The authors should check the space and typing through the manuscript such as line 19, 64,78, 144, 163-165 and so on.

2.     The ‘Abs’ of Figure 5 was missing

Reviewer 3 Report

This manuscript described some phosphorus Schiff base-Pd(II) complexes with DNA binding and anticancer potential. Two compounds were screened for anticancer activity against different cancer cell lines and were found to be highly cytotoxic, and competitive with the anticancer agent cisplatin. Nevertheless, the reviewer recommends minor revision of this manuscript before publishing in Pharmaceutics for the following reasons:

1.      For the Schiff base ligands synthesis, only 1HNMR and 31PNMR were given. 13CNMR for each compound should be added before publishing.

2.      In order to test the DNA binding ability, only B1 and B2 were subjected to experimental tests but omitted the other 6 compounds. These results lack solid evidence even though docking was done for all of them. Please add the binding results of the rest complexes.

3.      Confusing title labeling: for example following section two they subheaded with 1.9..and  section three starts with 1.2….Please check whether some part was missing or just mislabeled.

4.      Don’t exaggerate the conclusion based on current research typically in the conclusion part! Please evaluate your results properly.

Reviewer 4 Report

High number of papers have been continuously published on new Pd(II)-containing complexes (many of them as models to the corresponding Pt(II)-ones). In this subject, this paper summarizes results on a well-done work.  As a consequence,  although the novelty is moderate, its publication can be suggested.

Author Response

We thank to the reviewer for their comment and suggesting the publication of our manuscript.

Round 2

Reviewer 2 Report

There were reasonable and adequate responses on the revised verson.